# Prospects for the Use of Cannabinoids in Oncology and Palliative Care Practice: A Review of the Evidence

**DOI:** 10.3390/cancers11020129

**Published:** 2019-01-22

**Authors:** Tomasz Dzierżanowski

**Affiliations:** Laboratory of Palliative Medicine, Department of Social Medicine and Public Health, Medical University of Warsaw, ul. Oczki 3, 02-007 Warszawa, Poland; tomasz@adiutus.pl or tdzierzanowski@wum.edu.pl; Tel.: +48-601-334-001

**Keywords:** cannabis, cannabinoids, cancer, palliative care

## Abstract

There is an increased interest in the use of cannabinoids in the treatment of symptoms in cancer and palliative care patients. Their multimodal action, in spite of limited efficacy, may make them an attractive alternative, particularly in patients with multiple concomitant symptoms of mild and moderate intensity. There is evidence to indicate cannabis in the treatment of pain, spasticity, seizures, sleep disorders, nausea and vomiting, and Tourette syndrome. Although the effectiveness of cannabinoids is limited, it was confirmed in neuropathic pain management and combination with opioids. A relatively favorable adverse effects profile, including no depressive effect on the respiratory system, may make cannabis complement a rather narrow armamentarium that is in the disposition of a palliative care professional.

## 1. Introduction

Marijuana and hashish are frequently used psychoactive substances. However, they have also been used for medical purposes for thousands of years. There is an increased interest to use cannabinoids in the treatment of symptoms in patients with cancer or HIV, in Tourette syndrome, epilepsy, spasticity, and in digestive disorders [1,2]. Controversies around the legalization of *cannabis* for recreational use impede the approval of its medical preparations. They recall those of the ’80s that impeded implementation of cancer pain treatment with opioids and tend to express political, rather than medical positions. There are two contradictory positions regarding the medical use of cannabis. One is affirmative and even irrespective of the clinical evidence. The second one is conservative with prejudices and fears. The right approach should be evidence-based. In this light, there are critical questions regarding the medical use of cannabis. Is it an effective and safe symptom controlling medicine in palliative care patients? Does it have anti-cancer life-prolonging properties? In which indications has cannabis appeared useful? How much can we expect from cannabis in the management of pain and other symptoms? What should a palliative care physician and an oncologist know about cannabis and cannabinoids? This paper aims to summarize the theoretical and clinical rationale for the use of cannabinoids in the treatment of palliative care patients.

## 2. Endocannabinoid System

The cannabinoid system consists of two main cannabinoid receptors CB_1_ and CB_2_, and their endogenic ligands. CB_1_ receptors were discovered in 1988, and two years later their responsiveness to Δ9-tetrahydrocannabinol (Δ9-THC) was confirmed. Δ9-THC is the main psychoactive constituent of marijuana—the product of the dried flowers and subtending leaves and stems of the female *Cannabis* spp. plant, which their name derives from [3]. CB_1_ are G protein-coupled receptors (GPCR), which once activated, inhibit adenyl cyclase and production of cAMP. As a consequence, neuronal voltage-dependent calcium currents close and potassium currents open, which lead to hyperpolarization of the neuron and inhibition of transmission of an electric impulse. The selectivity of the agonists is small, unlike the antagonists.

### 2.1. Receptors

CB_1_ receptors are present in all the central and peripheral nervous systems, mainly on axons, but also on neuronal cell bodies and dendrites. Their highest concentration is found in cortical structures including the hippocampal formation and the olfactory bulb. A less dense dispersion is found in the basal ganglia, the cerebellum, and the spinal cord. This explains the effects of cannabinoids on memory, emotion, cognition, smell distortion and movement and pain transmission. They are also especially concentrated in the periaqueductal grey matter (PAG), *nucleus tractus solitarius*, the ventral posterolateral nucleus of the thalamus and the dorsal horn of the spinal cord that are involved in the modulation of nociceptive transmission. Part of these neurons are GABAergic, which may elucidate cannabinoids’ role in modulating GABAergic neurons [4,5,6]. CB_1_ receptors are also distributed in peripheral tissues: liver, pancreas, white adipose tissue and skeletal muscles, which may be linked to the metabolic consequences of CB_1_ receptor antagonism [7,8,9,10].

Large diameter primary afferent Aβ and Aδ fibers are more densely populated with CB_1_ receptors than with μ opioid receptors, which suggests possible effectiveness in suppressing neuropathic pain caused by nerve dissection [11]. On the other hand, CB_1_ receptors are sparse in the brainstem, which may explain the neutral impact of their ligands on the respiratory system [4,12].

CB_2_ receptors are similar to CB_1_ in their amino acid structure in 44% and are present peripherally on some immune cells derived from macrophages (such as microglia, osteoclasts, and osteoblasts), on neurons, and also in some solid tumors, which suggests potential anti-cancer activity [13,14,15].

The major cannabinoids have multiple targets within the central nervous system and can modulate the activity of neurons, glia, and microglia, and it is unknown which mechanisms are critical for their anti-seizure effect [16].

### 2.2. Ligands and Their Mode of Action

Around 480 cannabinoid receptors’ ligands have been isolated, including over 110 produced by *Cannabis sativa*, composed of different chemical structures. The cannabinoids may be grouped in:Endogenic cannabinoids—eicosanoids derived from arachidonic acid; e.g., anandamide (AEA; from Sanscrit *Ananda*—internal bliss)—partial agonist, 2-arachidonoylglycerol, palmitoylethanolamide),Phytocannabinoids (classical cannabinoids), e.g., Δ9-tetrahydrocannabinol (Δ9-THC), cannabidiol (CBD), cannabinol, cannabicyclol, 11-hydroxy-Δ8-THC-dimethylheptyl,Synthetic cannabinoids: CP55940; indols (e.g., WIN55212); antagonists and reverse agonists (SR141716A and AM251 for CB_1_, SR145528 and AM630 for CB_2_).

Plenty of other ligands act primarily on non-cannabinoid receptors with some activity on CB_1_, e.g., N-arachidonoyldopamine (NADA) which is primarily a vanilloid receptor agonist, but expresses some activity on CB_1_ [17].

The activity of cannabinoids is not limited to CB_1_ and CB_2_. They interact with multiple receptors and systems, including GABA-ergic/glutaminergic, noradrenergic, and opioid systems. Antinociception induced by cannabinoids is also in part an effect of the release of norepinephrine in descending inhibiting pathways, and their action synergistic to opioids. They express agonism to transient-receptor potential vanilloid 1 (TRPV1), which is important in inhibiting of thermal and mechanical stimuli, hyperalgesia and allodynia. There are some different receptors susceptible to cannabinoids, responsible for different effects, such as:Calcitonin gene-related peptide (CGRP)—vasodilatation,Calcium channels,Peroxisome proliferator-activated receptor gamma (PPARγ)—expressed mainly in the adipose tissue, in a minor degree in kidneys, heart, and lungs; affect differentiation and maturity of adipocytes,Nuclear factor kappa-light-chain-enhancer of activated B-cells (NF-κB)—plays a key role in the regulation of immunologic response to infection; downregulation of NF-κB is linked to carcinogenesis, inflammation and autoimmune diseases,G protein-coupled receptor 18 (GPR18)—responsible for induction of apoptosis of proinflammatory macrophages.

Cannabinoids express non-receptor activity, such as inhibition of cyclooxygenase (more COX-2 than COX-1), which reflects their anti-inflammatory action. On the other hand, classical analgesics like non-steroid anti-inflammatory drugs, opioids, and paracetamol, as well as antidepressants, increase the activity of the endocannabinoid system [18].

## 3. Phytocannabinoids

There are many natural molecules active versus CB_1_ and CB_2_. The main source of phytocannabinoids is hemp (*Cannabis*). By 2009 there were 108 cannabinoids isolated from *Cannabis sativa*, and classified into ten types and 14 subtypes [19]. Cannabis is an annual, dioecious, flowering, fast growing herb, which leaves palmately compound or digitate, with serrate leaflets [20].

There are three main taxons:(1)*Cannabis sativa* (*C. sativa subsp. sativa, C. sativa var. vulgaris*)—a nominative subspecies, 150–250 cm tall, spread out in different regions of the world,(2)*C. ruderalis* (*C. sativa var. spontaneous*)—identified within the nominative subspecies, up to 150 cm tall, growing in Central Asia, containing a small amount of cannabinoids,(3)*C. indica* (*C. sativa subsp. indica*)—reaching 300–350 cm of height, growing primarily in India, Afganistan and Iran, and containing the highest amounts of THC and CBD.

The taxonomy of *Cannabis* spp. is still being discussed. There is also the chemotaxonomic classification of *Cannabis* spp., dependent on the content of cannabinoids [21]. Cannabinoids are also present in other plants, like *Echinacea purpurea*, *E. angustifolia*, *Helichrysum umbraculigerum*, or black truffles [22].

## 4. Terminology of Cannabis

The European Monitoring Centre for Drugs and Drug Addiction (EMCDDA) proposed the use of the following names:Hemp—cannabis of the fiber-phenotype; contains less than 0.3% THC,Herbal cannabis (i.e., ‘marijuana’, ‘leaf’, ‘weed’, ‘grass’ etc.)—the fresh or (more commonly) dried leaves and flowering tops, but excluding stalk, roots, and seeds of Cannabis sativa; contains 0.5–5% of THC; usually inhaled in a form of a cigarette (‘joint’), often mixed with tobacco,Cannabis resin (‘hashish’)—compressed resin, usually mixed with herbal cannabis, tobacco or another type of herb, to allow it burn in the form of a joint; contains 2–20% of THC,Hash oil or cannabis oil—solvent extracts of herbal cannabis or cannabis resin (usually 10–30% of THC) [23].

## 5. Routes of Administration

In ancient cultures, the consumption of cannabis had a religious, shamanic, or spiritual context, dating back to more than 2000 years BC. In some traditions, cannabis plays the same social role as tobacco, tea or coffee. An infusion called ganja (from Hindi and Urdu: *gānjā*) was predominant in Jamaica.

Smoking is the most frequent way of cannabis consumption, also for medical purposes. Cannabis may be heated to 175–225 °C, to allow for inhalation of the evaporated substances without burning. Vaporizers are more advantageous to joints, water pipes, and filters, as heating marijuana to 185 °C prevents against producing carcinogenic benzene, toluene and naphthalene [24]. Oral administration is possible too, however with four-fold less bioavailability of cannabinoids. It has been the most often route of administration of cannabis for medical purpose for ages.

## 6. Pharmacokinetics and Metabolism

When inhaled in the form of a cigarette (*joint*) 20–45% of THC is absorbed. The maximum brain concentration appears in 15 min and corresponds to the time to maximum psychic and somatic effects that last over the next 2–4 h (*plateau*) and then gradually cease.

THC is highly lipophilic, which accounts for its significantly large distribution volume, cumulation in fat-rich tissues (brain, lungs, kidneys), and slow elimination [25]. This high lipophilicity and poor solubility in water limits use in infusions, although 15% of THC and tetrahydrocannabinol acid (a derivative with some immune-modulating properties and bereft of psychoactivity) comes to the solution during infusion preparation [26,27].

After oral administration, the total bioavailability of THC varies and reaches 10–20% and is 10–30% of the dose that would be absorbed when inhaled. THC degrades in the stomach and intestines, which decreases its bioavailability. Only 6 ± 3% out of 20 milligrams of THC contained in a cake reaches the central compartment and is significantly variable in individuals [28]. The psychoactive effect develops more slowly than after inhalation. It appears in 30–120 min with peak intensity after 60–120 min, and it maintains longer (5–12 h). The blood concentration of much more psychoactive 11-OH-Δ9-THC is much higher than after smoking [29].

The action after sublingual spray administration starts after 15–45 min and lasts similarly to the oral forms (6–8 h) [30]. Due to variable pharmacokinetics, it is considerably more difficult to set an appropriate dose of an oral formulation than of an inhaled one. 

Metabolism of the cannabinoids depends mainly on their route of administration. After oral intake, Δ9-THC is mainly metabolized in the liver by cytochrome P450 (CYP) 2C9, 2C19 and 3A4 to much more psychoactive 11-hydroxy-Δ9-THC and inactive 11-nor-9-carboxy-Δ9-THC [31].

## 7. Adverse Effects

Cannabis taken sporadically produces multiple psychoactive effects. In most people, THC causes euphoria, muscle relaxation, and intensification of sensory feelings (e.g., palate, thus it impacts an appetite). However, in some individuals, anxiety and panic reactions appear instead of euphoria. Sensory distortions are frequent. Psychomotor, cognitive and behavioral disorders seem to be dose-dependent.

There is a systematic review of 3695 reports on toxicity associated with synthetic cannabinoids. They include physiological (e.g., tachycardia, hypertension, nausea/vomiting), emotional (e.g., agitation, irritability, paranoia), behavioral (e.g., drowsiness, aggression) and perceptual (e.g., hallucinations) symptoms. The most frequent are tachycardia (30.2% of cases), agitation (13.5%), drowsiness (12.3%), nausea or vomiting (8.2%) and hallucinations (7.6%). Death or serious adverse effects were rare (e.g., death 0.2%, stroke 0.1%, myocardial infarction 0.09%) [32]. These figures are coherent with the analysis of 256 reports with around 4000 cases and 26 deaths. Most presentations were mild or moderate, typically involved young males with tachycardia (37–77%), agitation (16–41%) and nausea (13–94%) requiring only symptomatic care with a length of stay of fewer than eight hours [33]. It should be underlined that these statistics refer to uncontrolled recreational consumption of synthetic cannabinoids.

Unpleasant and sometimes serious ailments are depersonalization, paranoia, distortion of time perception and anticholinergic effect (dry-mouth, double-vision, urinary retention, decreased heart rate, anhydrosis, increased body temperature). THC also causes orthostatic hypotony and reflex tachycardia up to three hours after dose intake. Ataxia and decreased muscle strength have also been reported. Sporadic marijuana use is not usually dangerous, although the withdrawal syndrome may entail severe depressive disorders with suicidal thoughts as well [25].

There are reports on the increased risk of systolic hypertension, ischemic stroke, and ventricular arrhythmias. On the other hand, the repeated intake of marijuana may induce tolerance of its cardiovascular effects, through the decrease in the number of receptors or their susceptibility [34,35].

Cannabis affects the immune system, causing cellular and humoral response deficiency.

There is an increased risk of chronic bronchitis and cancer, although:In the Coronary Artery Risk Development in Young Adults (CARDIA) study the occasional use of marijuana was not related to any respiratory system adverse events, even when taken over 20 years [36],The risk of cancer in long-term or addicted smokers of marijuana does not increase, but side effects are possible after large intake [37],Smoking of a mix of tobacco and marijuana has a more negative impact than marijuana alone [38].

The psycho-nervous symptoms include deterioration of memory and attention, depletion of organization and integration of complex information, disorientation, amnesia, illusions, hallucinations, anxiety, and arousal, exacerbation of schizophrenia.

The reproductive system may be affected as well by a decreased testosterone concentration, and consecutive depletion of number, mobility, and vitality of sperm cells. The regular use of marijuana entails the ovarian cycle disorders and low birth mass in newborns of marijuana addicted mothers.

It should be emphasized that the symptoms discussed above are the effect of uncontrolled and risky dosing of cannabis, which is not the case of medical use under the guidance and control of a physician. As medicines, cannabinoids have a much better safety profile than many other substances used in oncology and palliative care. Lethal overdosing is impossible, as CB_1_ receptors are sparse in the brainstem cardiovascular and respiratory centers [30]. In the elderly, the therapeutic use of cannabis is safe and efficacious and may decrease the use of other prescription medicines, including opioids [39].

The adverse effects of cannabis used for medical purposes are frequent but mostly mild and do not require treatment. They include disorientation, dizziness, euphoria, confusion, dry mouth, nausea, somnolence, loss of balance, fatigue, weakness, drowsiness, hallucination, paranoia, anxiety [1]. In 14% of patients receiving a transmucosal spray (Sativex^®^, GW Pharma Ltd., Salisbury, Wiltshire, UK), there were local adverse events in the site of application reported, such as glossodynia, mouth ulceration, oral discomfort, oral pain, application site irritation, application site pain, pharyngitis, throat irritation and dysgeusia [40].

In a recently published systematic review of systematic reviews reporting adverse events of medical cannabinoids, overall adverse events were statistically significant, with numbers needed to harm (NNH) of five to eight. No systematic review demonstrated serious adverse events rates which were statistically different from placebo. However, it has been recognized that the rate of adverse events is probably underreported, as many studies enrolled experienced cannabis users, who have a reduced risk of adverse events, while some serious events like psychosis appear to be more common among naïve users. What more, some adverse events have a greater magnitude of effect than the potential benefits for the conditions targeted [41].

It is important to emphasize, that many RCTs were short in duration, so long term safety and frequency of rare serious adverse effects of cannabinoids remain undetermined.

## 8. The Risk of Physical Dependence and Psychic Addiction

Long-term marijuana use may lead to physical dependence, and more quickly than tobacco and alcohol do. Physical dependence is most often manifested in the appearance of withdrawal symptoms when cannabis use is abruptly halted or discontinued. Withdrawal symptoms appear within the first one to two days following discontinuation and reach peak intensity between days two and six, with most symptoms resolving within one to two weeks. The most common symptoms include cravings, anger or aggression, irritability, anxiety, nightmares/strange dreams, insomnia/sleep difficulties, headache, restlessness, and decreased appetite or weight loss [25,42,43].

The transition to cannabis addiction occurs considerably faster than the transition to nicotine or alcohol addiction. However, the cumulative probability estimate of transition from use to addiction was 67.5% for nicotine users, 22.7% for alcohol users, 20.9% for cocaine users, and only 8.9% for cannabis users [44]. The risk of psychic addiction is higher in adolescents than in adults. It occurs in one in six users if regular use of marijuana commences in their teens while occurring only in one in ten if started during adulthood [45,46]. The regular consumption of marijuana may be especially disadvantageous in young people, in whom there is an increased risk of hard drugs addiction [25].

On the other hand, the pre-clinical studies suggest that CBD may have therapeutic potential for the treatment of opioid, cocaine and psychostimulant addiction, and may also be beneficial in cannabis and tobacco addiction in humans [40].

## 9. Drug Interactions

Cumulation of cannabinoids in tissues rich in fat increases the risk of interactions, with such medicines as opioids, benzodiazepines, phenothiazine, beta-adrenolytics, anticholinergic agents, barbiturates, and cholinesterase inhibitors. 

Cannabinoids are weak inhibitors of cytochrome P450 (CYP). Δ9-THC inhibits CYP3A4, 3A5, 2C9 and 2C19. CBD inhibits CYP2C19, 3A4 and 3A5, although this can be observed only at doses higher than clinically used. Nevertheless, caution is advised with concomitant fentanyl and amitriptyline use, as both are metabolized through CYP3A4, and 2C19 [47].

Most drug interactions are an effect of the concurrent use of other agents’ depression of the central nervous system. Clinically significant interactions are rare, and cannabis may be combined practically with any medicine [30].

## 10. Medical Forms of Cannabis

Cannabinoid formulations used for medical purposes may be divided into non-standardized products and those with a standardized content of active substances. The non-standardized forms are simply marijuana, herbal cannabis, resin or oil, from legal or unofficial sources, dependent on local regulations and restrictions. Increasing public awareness of the possible medical use of cannabis and pressure on authorities has led to attenuation of highly restrictive laws or even abolishment of any restrictions against cultivation and distribution of cannabis for medical use. There is also an increasing number of medical products. There is a very narrow armamentarium available to palliative or supportive care specialists. Any novel medicine that adds value to the currently available treatment would be appreciated. Cannabinoids seem to have limited effectiveness in the treatment of pain, nausea and vomiting, spasticity, seizures, and mood disorders. Regarding pain treatment, they might be considered as an adjuvant to opioid therapy, but also (in less severe cases) before opioids. In many cases, a moderate dose of cannabinoids is sufficient and without the negative effects, that opioids bear.

Access to herbal cannabis for medical purposes varies in different regions. In some countries (e.g., Croatia, Denmark, Finland, Norway, Serbia, Switzerland, Sweden) programs are allowing an authorized physician to prescribe herbal cannabis. In some countries, the access is broader (e.g., Czechia, Germany, Holland, Israel, Italy, and San Marino). The registered indications differ too, dependent on the country: pain with spasticity, persistent pain refractory to the conventional therapy, management of chemotherapy and radiotherapy-induced nausea and vomiting, in palliative care, in HIV/AIDS patients, or multiple sclerosis [48].

The standardized dried herbal cannabis differs in content and the ratio of THC (from <1% to 22%) and CBD (1–9%), which suggests various indications, as the clinical effects of THC and CBD differ (Table 1) In general, THC is responsible for euphoria, relaxation, and stimulation of appetite. CBD has anxiolytic, anti-depressant, anti-convulsant, and no psychoactive effects. It also prevents the pro-psychotic actions of THC and decreases appetite. Both cannabinoids bring pain relief. Thus, the choice of a form with higher THC or CBD concentration depends on the specific clinical situation. 

Dronabinol is available as 2.5 mg and 5 mg gel capsules containing THC in sesame oil and is registered in the USA for the treatment of AIDS-related anorexia associated with weight loss and severe nausea and vomiting associated with cancer chemotherapy, under the Controlled Drugs and Substances Act [49].

Nabilon, a synthetic THC analog, is available as 0.25, 0.5 and 1 mg tablets approved by the FDA for the treatment of nausea and vomiting associated with cancer chemotherapy in patients who have failed to respond adequately to conventional antiemetic treatments. The use is restricted because a substantial proportion of any group of patients can be expected to experience disturbing psychotomimetic reactions not observed with other antiemetic agents. As there is potential to alter the mental state, close supervision of the patient by a responsible individual particularly during initialization of the therapy and during dose adjustments is required [50].

Nabiximols is a whole plant extract of cannabis containing THC and CBD in a ratio of 1:1, available in some countries as an oromucosal spray (e.g., Sativex^®^ 2.7 mg THC and 2.5 mg CBD in one dose). The product is indicated as an adjunctive treatment for symptomatic relief of spasticity in adult patients with multiple sclerosis who have not responded adequately to other therapy and who demonstrate meaningful improvement during an initial trial of therapy. It may also be useful as adjunctive treatment for the relief of neuropathic pain in adult patients with multiple sclerosis, and as an adjunctive analgesic treatment in adult patients with advanced cancer who experience moderate to severe pain during the highest tolerated dose of strong opioid therapy for persistent background pain [47].

The main health concern regarding the medical use of cannabis is the respiratory consequences of smoking [24]. During smoking, more than 2000 compounds may be produced by pyrolysis [31]. The non-pyrolytic vaporization reduces the formation of hazardous combustion carcinogenic products, such as tar, polycyclic aromatic hydrocarbons (PAH), carbon monoxide, and other carcinogens (e.g., benzene) [51]. While around 150 chemicals are identified in the smoke of combusted cannabis, among them five PAHs, known as strong carcinogens, only three are present in the vapor [52,53].

Vaporizing is more efficient than smoking because a part of THC in marijuana cigarettes is destroyed by pyrolysis during smoking. Inhalation by vaporization is a promising application mode for cannabis in medicine, and are the alternative to waterpipes and solid filters. Electrically-driven vaporizers decarboxylate cannabinoid acids at about 200 °C and release neutral, volatile cannabinoids, which enter the systemic circulation via pulmonary absorption from the vapor. However, the release of cannabinoids into the vapor is dependent on the device used and varies from 48.5% to 82.7%. Vaporizers can also be used to inhale cannabis oil and waxy extracts from the plant. 

## 11. The Medical Indications for Cannabinoids

Although there are few good quality randomized clinical trials (RCTs), there are 126 items in the PubMed having “systematic review” and “cannabinoids” (or “cannabis”) in the title. Half of them were published in the last three years, which indicates an increasing interest in the medical use of cannabis. Both favorable, as well as unfavorable results, should be treated cautiously. The additional difficulty is born by the fact that some of the data refer to inhaled cannabis when the other ones to the standardized medications in the form of tablets or extracts.

### 11.1. Spasticity

Brain spasticity, spasticity due to spinal cord injury or multiple sclerosis (MS) is one of the first clinically confirmed indications. It is raised, however, that the cannabinoids may also induce or exacerbate such symptoms as spasticity, ataxia or muscle debilitation, and the adverse effects are very frequent. Nevertheless, in a questionnaire-based survey, over 87% of the responders with MS reported improvement of spasticity during sleep, when waking or walking, as well as decreased pain and tremor [54,55].

There are fifteen systematic reviews assessing cannabinoids in the treatment of spasticity or spasticity-related pain by the submission of this paper (December 2018), and six of them appeared in 2018.

In the meta-analysis of five RCTs, a significant reduction in the Ashworth spasticity scale has been confirmed in comparison to placebo [1].

In the meta-analysis of 16 trials in multiple sclerosis and paraplegia, moderate-certainty evidence suggested a non-statistically significant decrease in spasticity, and spasm frequency [56].

In an Italian systematic review with the meta-analysis including 15 trials of cannabis compared with placebo in patients with multiple sclerosis, confidence in the estimate was high in favor of cannabis for spasticity (numerical rating scale and visual analogue scale, but not the Ashworth scale) and pain [57].

In the recent meta-analysis of 23 trials including 2270 patients on efficacy and safety of treatments for spasticity caused by multiple sclerosis, cannabinoids had shown a significantly better efficacy than placebo in the percentage of improved patients, but no significant difference was found in spasticity scale [58]. This could possibly be explained by the diversified clinical effects of cannabis, or merely average improvement versus placebo. There was also conclusive or substantial evidence that cannabis or cannabinoids are effective for the treatment of spasticity associated with multiple sclerosis in the update from the National Academies of Sciences, Engineering, and Medicine report on the therapeutic effects of cannabis and cannabinoids [59].

The plentitude of the secondary data resulted in a systematic review of reviews, in which the authors identified 11 eligible systematic reviews that are providing data from 32 studies, including ten moderate- to high-quality RCTs. Five reviews concluded that there was sufficient evidence that cannabinoids may be effective for symptoms of pain or spasticity in MS [60].

### 11.2. Pain

The analgesic effectiveness of cannabinoids is comparable to the weak opioids. There are few comparative studies though. Meaningful relief in moderate pain appears after a minimum 15–20 mg of THC, reaches a maximum in 3 h, and lasts up to 6 h, which suggests that THC should be administered every 6 h. 20 mg of THC is equianalgesic to roughly 120 mg of codeine. 

In the meta-analysis of eight trials, cannabinoids brought a mere reduction of pain vs. placebo (37% vs. 31%). However, the odds ratio for ≥ 30% pain relief (including cancer pain) vs. placebo is 1.41 and not statistically significant (0.99–2.00 95% CI) [1].

There are over 50 systematic reviews on cannabis preparations in the management of different pain conditions, and over 30 of them were published in the last three years (2016–2018). Some of them are of poor quality and thus will not be considered below.

In a systematic review of cannabinoids in cancer pain treatment, eight RCTs met the inclusion criteria. Low-quality evidence supported that cannabinoids, especially nabiximols, were effective analgesics for cancer pain. Few significant side effects or adverse reactions were reported, mostly cognitive changes and dizziness. There was low-quality evidence against THC as an effective analgesic for cancer pain, and pain relief was achieved only at high doses. However, significant cognitive impairment and dizziness limited the use of THC at these doses. An oral synthetic nitrogen analog of THC 4 mg (NIB) and oral benzopyranoperidine (BPP) 2–4 mg appeared to not be useful analgesics for cancer pain, although the quality of the evidence was low for both. Even when the analgesic effects from NIB were apparent, the frequency and severity of side effects made the medicine useless. Pain intensity worsened in patients administered BPP [61].

Nevertheless, different mechanisms, additional clinical benefits and no depressive effect on the respiratory system, allow for combined therapy with opioids. This synergetic effect of cannabinoids in combination with opioids in pain relief as well as chemotherapy-induced peripheral neuropathy (CIPN) was demonstrated in animal models. The analgesic effect of THC is, at least in part, mediated through delta and kappa opioid receptors, indicating an intimate connection between cannabinoid and opioid signaling pathways in the modulation of pain perception [62]. In conclusion of the review above, there was low evidence that cannabinoids were effective adjuvants for cancer pain not completely relieved by opioid therapy, and they appeared to be safe in low and medium doses [61]. 

In two RCTs on cannabinoids in adults with moderate to severe cancer pain, currently using opioids, the results appeared inconsistent. In the first one, with nabiximols (THC:CBD ratio close to 1:1) administered in the form of a transmucosal spray, there was a statistically significant difference of nabiximols versus placebo, and no difference versus THC, in pain relief in the individuals, who did not reach sufficient analgesia with opioids. The response (≥30% pain relief) rate after two weeks was 43% for nabiximols and 21% for placebo [63].

In the second RCT, different doses of nabiximols were compared to placebo in patients treated with stable doses of opioids. The effectiveness of low (1–4 sprays/day) and medium doses (6–10 sprays/day), but not at a high dose (11–16 sprays/day) was demonstrated. The adverse effects were dose-dependent, and statistically more frequent only at high doses of cannabinoids. The dose necessary to attain significant pain relief was around ten sprays, that is 27 mg of THC and 25 mg of CBD, which was near the maximum allowed daily dose (12 sprays/day—32 mg THC/30 mg CBD) [64]. 

In the International Association for Study of Pain (IASP) Guidelines, there are weak recommendations against cannabinoids for the treatment of the neuropathic pain, due to inconclusive data, in spite of strong theoretical premises for such action [65]. Nevertheless, in a meta-analysis of 6 RCTs on neuropathic pain, cannabinoids appeared more effective than placebo (odds ratio 1.38, 95% CI: 0.93–2.03 [1].

In a recent German systematic review with meta-analysis, nine studies at moderate risk of bias, with a total of 1561 participants, were included. The quality of evidence was rated according to GRADE as low or very low. In cancer patients, there were no significant differences between cannabinoids and placebo for ≥30% decrease in pain. There were no differences between cannabinoids and placebo in symptoms of dizziness or poor mental health [66].

The results of the recent systematic review with meta-analysis of cannabis and cannabinoids for the treatment of adult patients with chronic non-cancer pain, with 9958 participants in 47 RCTs and 57 observational studies, appeared disappointing. There was no significant difference for 50% reduction in pain vs. placebo, and for 30% reduction in pain, the NNT values were very high (24; 15–61 95% CI), while numbers needed to harm was relatively low (6; 5–8 95% CI). In conclusion, the evidence for the effectiveness of cannabinoids in chronic non-cancer pain is unfavorable, and it seems unlikely that cannabinoids are highly effective medicines in this indication [67].

### 11.3. Nausea and Vomiting

The endocannabinoid system has shown an important role in the regulation of nausea in preclinical studies. Smoked or orally administered THC appeared to be effective in reducing chemotherapy-induced vomiting and nausea (CINV) in several trials [68].

There are over 20 systematic reviews in which cannabinoids were assessed regarding nausea and vomiting. In the oldest one (2001), they appeared effective in the prevention and treatment of CINV in comparison to placebo and not worse than the antiemetics, such as prochlorperazine, metoclopramide, chlorpromazine, thiethylperazine, haloperidol, domperidone, or alizapride. The number needed to treat (NNT) was six for complete control of nausea and eight for complete control of vomiting [69]. In another systematic review of anti-emetic management, cannabinoids were effective for nausea and vomiting in people receiving chemotherapy but were associated with a high and often unacceptable burden of adverse effects [70].

Another meta-analysis showed that cannabinoids were associated with a greater average number of patients showing complete nausea and vomiting response (47% vs. 20%; odds ratio 3.82 (95% CI, 1.55–9.42)) [1].

In a quite recent systematic review of systematic reviews of efficacy, tolerability, and safety of cannabinoids for CINV, six systematic reviews of RCTs with dronabinol, levonantradol, nabilone, and nabiximols were analyzed. There was moderate quality evidence that cannabinoids are effective, but less tolerated and less safe compared to placebo and conventional antiemetics for CINV. The authors conclude that with safe and effective antiemetics available, cannabinoids cannot be recommended as first- or second-line therapy for CINV [71]. In a similar newer review of systematic reviews, cannabinoids seemed to be more effective than placebo, equal to prochlorperazine for reducing CINV, and to be preferred by patients. According to the authors, although there is no good quality evidence to recommend or not the use of them for CINV, they represent a valuable option for treating CINV, despite the adverse events related to treatment [72].

There is sparse evidence for the effectiveness of cannabinoids for the treatment of non-chemotherapy associated nausea and vomiting [73].

In one of the most recent systematic reviews, there were no significant differences between cannabinoids and placebo for improving nausea and vomiting in adult palliative care cancer and HIV patients [66]. It should be kept in mind, that cannabinoids may induce nausea or vomiting by themselves, and these are ones of the most frequent side effects [32,33].

### 11.4. Seizures

Seizures are a frequent problem in cancer patients, especially in those with metastases to the brain. 30% of epileptic patients have refractory seizures [74]. Preclinical and preliminary data from studies in humans suggest that cannabidiol and Δ9-THC may be effective in the treatment of some patients with epilepsy [75]. Despite this empiric evidence, the mechanisms by which they exert anti-seizure effects are poorly understood [16]. Three high-quality RCTs were completed recently, and the evidence for their efficacy in the refractory epilepsy is solid [76,77]. They reduce generalized, focal and absence seizures. On the other hand, Δ9-THC or synthetic cannabinoid agonists can provoke or exacerbate seizures or interact with other drugs. However, these side effects do not outweigh the overall benefit of the drugs [78].

### 11.5. Sleep Disorders

Nabiximols and herbal cannabis improve quality of sleep and rest, and they attenuate insomnia and sleep apnea index [1].

Cannabidiol (CBD) may have therapeutic potential for the treatment of insomnia, while Δ9-THC may decrease sleep latency but could impair sleep quality long-term. Synthetic cannabinoids (e.g., nabilone, dronabinol) may have a short-term benefit for obstructive sleep apnea due to their modulatory effects on serotonin-mediated apneas. CBD may be possibly effective for REM sleep behavior disorder and excessive daytime sleepiness, while nabilone may reduce nightmares associated with posttraumatic stress disorder and may improve sleep among patients with chronic pain [79]. The evidence that cannabinoids are associated with an improvement in sleep quality or disturbance in palliative care patients is contradictory [1,66]. It seems that the effects are dose-dependent. The improvement is achieved at low doses only [64].

### 11.6. Appetite

One study reported a significantly greater increase in appetite among patients with AIDS who received cannabinoids. Dronabinol was associated with increased appetite above baseline (38% vs. 8% for placebo, *p* = 0.015), and weight was stable in dronabinol patients, while placebo recipients had a mean loss of 0.4 kg (*p* = 0.14) [80]. The quality of evidence was very low [66,81,82].

Dronabinol appeared less effective than megestrol acetate. No improvement in appetite nor body weight gain was achieved by adding dronabinol to megestrol acetate either [83].

In a randomized, double-blind, placebo-controlled pilot trial, THC-treated cancer patients reported improved and enhanced chemosensory perception and food ‘tasted better.’ Premeal appetite and proportion of calories consumed as protein increased compared with placebo [84].

### 11.7. Mood and Psychotic Disorders

In multiple sclerosis patients, cannabis brings improvement of depression and anxiety [54]. Cannabinoids have anxiolytic, sedative and a soporific activity, which may be valuable, as many patients with advanced cancer tend to have adaptation disorders, such as depression or anxiety. Additionally, they attenuate alcohol or opioids withdrawal syndrome. The evidence is at high risk of bias though [1]. Comparative studies, however, have not demonstrated an antidepressant effect of cannabinoids, and one of these studies has even shown a pro-depressive effect when cannabinoids were administered in large doses [64].

### 11.8. Glaucoma

Cannabinoids effectively lower the intraocular pressure and have neuroprotective actions in animal models and patients [85]. However, cardiovascular and neurological effects may theoretically reduce the beneficial effect of lowering intraocular pressure by reducing ocular blood flow [86]. The evidence is rather against supporting the medical use of marijuana for the treatment of glaucoma due to the short duration of action, the incidence of undesirable psychotropic and other systemic side-effects, and the absence of scientific evidence showing a beneficial effect on the course of the disease [87].

### 11.9. Tourette Syndrome

THC significantly reduces the severity of tics and obsessive-compulsive disorder in patients with Tourette syndrome [88,89]. According to the updated report of the National Academies of Sciences, Engineering and Medicine (2018) from a review of current medical literature on the health effects of cannabis and cannabinoids, there was conclusive or substantial evidence that cannabis or cannabinoids are effective for the treatment of pain in adults, chemotherapy-induced nausea and vomiting and spasticity associated with multiple sclerosis. Moderate evidence was also found for the efficacy in secondary sleep disturbances. However, the evidence supporting improvement in appetite, Tourette syndrome, anxiety, posttraumatic stress disorder, cancer, irritable bowel syndrome, epilepsy and a variety of neurodegenerative disorders was described as limited, insufficient or absent [59]. 

## 12. Anti-Cancer Activity

Cannabinoids have demonstrated anti-cancer effects in different in vitro and in vivo models of cancer [90]. They induce cancer cell death by apoptosis and the inhibition of cancer cell proliferation, angiogenesis, invasion, and metastasis [91]. In a pilot phase I clinical study, patients with actively-growing recurrent glioblastoma, for whom standard therapy had previously failed, underwent intracranial THC administration. Although no statistically significant conclusions could be drawn from such a small cohort (9 patients), the results suggest that some patients might have responded to THC treatment in terms of a decreased tumor growth rate [92]. These findings encourage further investigations of the potential use of cannabinoids in cancer therapies. Two clinical trials in patients with recurrent glioblastoma multiforme and solid tumors are currently ongoing [91]. An in-depth presentation of particular anti-cancer mechanisms of *cannabis* exceeds the scope of this article.

It should be emphasized that there is no evidence for the clinical effectiveness of *cannabis* in the treatment or prevention of cancer so far, and the preliminary results do not allow to recommend, nor dissuade from its deployment in anti-cancer treatment. This is a frequent issue raised by patients and their families, and suggesting the use of cannabinoids as anti-cancer measures would only bear irrational hope.

## 13. Summary 

Cannabinoids, thanks to their multimodal activity and good safety profile may offer a valuable supplement to palliative treatment. Introduction of medical cannabis raises unnecessary and excessive controversies, similar to those in the 1980s referring to the use of morphine as the key strategy of strong cancer pain therapy. In the meantime, their safety profile is incomparably better than that of opioids. What’s more, in the states where medical cannabis laws have been introduced, the mortality due to unintentional overdosing of the opioid analgesics dropped by 25% [93]. The dispute should put aside the problem of the legalization of marijuana and hashish for recreational use, as it may allow misinterpretation of the efforts to better control suffering of palliative care patients with the liberalization of marijuana. The decisions should be taken upon the rational and temperate analysis of the evidence. There are more and more arguments supporting the accessibility and reimbursement of cannabis products for selected indications that were discussed in detail. The complex, though moderate, action of cannabis makes it suitable for the treatment of concomitant symptoms, such as pain, chemotherapy-induced nausea and vomiting, spasticity, seizures, mood disorders, loss of appetite, which is a frequent condition in the palliative care patient. That multimodal, although moderate, action of cannabis, may be sufficient to attain good symptom control and reduce the number of drugs used. In many cases, pain is mild to moderate, and cannabis might appear sufficiently effective, replacing more burdensome opioids as well. The use of cannabis as an adjuvant to the opioid analgesics seems also promising to overcome intractable pain. However, the uncertainties and controversies on the role and appropriate use of cannabis-based medicines still do not allow to recommend their use as a first-line treatment of chronic pain and other conditions, especially in primary care [94,95].

Overall, based on the current evidence, the benefits overweigh the possible risk for palliative care patients. However, it seems that they will be more meaningful in the earlier phases of progressive disease, particularly in active patients with less intense symptoms. Patients with refractory pain might benefit from their use as well.

In regions where the distribution of cannabis is illegal, some patients look for alternative measures for relieving of pain or other symptoms. Following the rule of the superiority of the patient’s interest, a physician should not restrain a patient from using cannabinoids for symptom control, on the stipulation that the product is pure cannabis. This position is a logical consequence of a relatively good safety profile, even when overdosing cannabis, and negligible risk of drug interactions. 

On the other hand, there is a threat that legalization of cannabis for medical use is only a pretext for increasing accessibility of it for recreational use. Good evidence for the medical use of cannabis is scarce. Further randomized controlled studies are necessary to confirm or redefine the role of cannabis in the treatment of palliative care patients.

## 14. Conclusions

(1) Cannabinoids, thanks to their multimodal activity and good safety profile may offer a valuable supplement to palliative treatment, and decrease the mortality due to overdosing of the opioid analgesics.

(2) The benefits seem to overweigh the possible risk for palliative care patients, particularly in the earlier phases of progressive disease, in patients with less intense symptoms, although patients with refractory pain might benefit from their use as well.

(3) There are more and more arguments supporting the accessibility of cannabis products for selected indications, although the evidence still do not allow to recommend their use as a first-line treatment of chronic pain and other conditions.

(4) There is lack of good quality evidence for the medical use of cannabis, and further randomized controlled studies are necessary to confirm or redefine the role of cannabis in the treatment of palliative care and cancer patients.

## Figures and Tables

**Table 1 cancers-11-00129-t001:** The differences in clinical effects of tetrahydrocannabinol (THC) and cannabidiol (CBD).

THC	CBD
psychoactive (euphoria or dysphoria, anxiety in some new users)	no psychoactive activity
relaxation and bliss	counteracts psychotropic effects of THC (short-term memory and cognitive disorders)
relieves pain	relieves pain
anti-inflammatory	anti-inflammatory
antispastic	anxiolytic and antidepressant
soporific	induces sleep, suppresses waking-up
stimulates appetite	suppresses appetite
	anticonvulsant
	possible anti-psychotic

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
