# Peer review of "Prospects for the Use of Cannabinoids in Oncology and Palliative Care Practice: A Review of the Evidence"

_cancers, 2019, doi:10.3390/cancers11020129_

Round 1

Reviewer 1 Report

Dear Author

I would like to thank you for great work done in putting all together the possible indications for cannabinoids' medical use along with short introduction regarding endocannabinoid system, routes of administration, forms of potential drugs and adverse effects.This work surely might me of a great value for physicians looking forward to use the medical cannabinoids in their practice- especially nowadays  when, in many countries, interest (both professional and lay) of cannabinoids is growing swiftly.

Whatsoever I would like to point some, in my opinion, very important isuues, regarding article:

Paragraph 7- Adverse effects

- lines 177-187: as for real purpose of the article I think it is resonable to use also data from RCTs. In a recently published systematic review of systematic reviews (Allan, GM. et all. 2018. Canadian Family Physician, 64, e78-e94.) the approximate risk of AEs reaches as high as 80% and its magnitude quite oftenly might overweight the benefits of treatments. The special risk group of patients are naive users (greater risk of psychosis) or elderly (neuropsychiatric or hypotensive effect). Up to date- as many RCTs were short in duration, we are unable to be sure about long term safety of cannabinoids (ie. undetermine prevalence of cannabinoid hyperemesis syndrome or amotivational issues)

- lines 219-220: as stated above- it is still not determine, and long term studies including especially naive patients are needed.

Paragraph 10- Medical forms of cannabis

- lines 264-266: I can not see any refernce that justify such strong statement; As far as I know- due to limited benefits and high risk of harm- cannabinoids could be considered in some specific situations as not more than third-line therapy.

Paragraph 11.1- Spasticity

- line 343: it should be noticed, that mentioned meta-analysis has shown a better efficacy of cannabinoids in the percentage of improved patients but no significant difference was found in spasticity scale. The author should point this and try to explain possibble causes of that.

Paragraph 13- Summary

- lines 532-540: I can not agree with the phrase, that there are more and more argument supporting cannabinoids use in different medical conditions. One of the most recent national guidelines (Canada; Allan, GM. et all, (2018). Simplified guideline for prescribing
medical cannabinoids in primary care. Canadian Family Physician, 64, 111–120.) in general recommends against use of cannabinoids with few excerption (some types of pain, CINV, and spasticity due to MS or SCI- use could be considered with considerations). 

- lines 538-540: it is not supported by evidence and national guidelines / recommendations as well as most recent EFIC (European Pain Federation) position paper on appropriate cannabinoids use.

Author Response

First of all, I would really thank you for your fair and detailed feedback. I have implemented most of your remarks. Please, find the detailed answers to your comments below.

Paragraph 7- Adverse effects

- lines 177-187: as for real purpose of the article I think it is resonable to use also data from RCTs. In a recently published systematic review of systematic reviews (Allan, GM. et all. 2018. Canadian Family Physician, 64, e78-e94.) the approximate risk of AEs reaches as high as 80% and its magnitude quite oftenly might overweight the benefits of treatments. The special risk group of patients are naive users (greater risk of psychosis) or elderly (neuropsychiatric or hypotensive effect). Up to date- as many RCTs were short in duration, we are unable to be sure about long term safety of cannabinoids (ie. undetermine prevalence of cannabinoid hyperemesis syndrome or amotivational issues)

Answer: I have considered this review (Allan 2018) and did not decide to cite it, because I found some discrepancies between the findings, the discussion and the summary. However, your remark on the special risk groups and the short duration of many RCTs is accurate, and I implemented it in lines 225-234. 

- lines 219-220: as stated above- it is still not determine, and long term studies including especially naive patients are needed.

Answer: you are right and I implemented it in lines 225-234.

Paragraph 10- Medical forms of cannabis

- lines 264-266: I can not see any refernce that justify such strong statement; As far as I know- due to limited benefits and high risk of harm- cannabinoids could be considered in some specific situations as not more than third-line therapy.

Answer: I thought over your remark and I must admit that you are right! I softened the statement in the lines 274-276.

Paragraph 11.1- Spasticity

- line 343: it should be noticed, that mentioned meta-analysis has shown a better efficacy of cannabinoids in the percentage of improved patients but no significant difference was found in spasticity scale. The author should point this and try to explain possibble causes of that.

Answer: I had big concerns towards this finding when writing the article. I have even posed a question to the author by ResearchGate, but I did not receive an answer yet. However, I tried to meet your expectations in lines 354-356, although I would skip these statements, if you agreed.

Paragraph 13- Summary

- lines 532-540: I can not agree with the phrase, that there are more and more argument supporting cannabinoids use in different medical conditions. One of the most recent national guidelines (Canada; Allan, GM. et all, (2018). Simplified guideline for prescribing

medical cannabinoids in primary care. Canadian Family Physician, 64, 111–120.) in general recommends against use of cannabinoids with few excerption (some types of pain, CINV, and spasticity due to MS or SCI- use could be considered with considerations).

Answer: see lines 541-542. Please note that the sentences refer to palliative care, not primary care. I have really a problem with the Canadian guidelines, as they are made and directed to the primary care physicians. I cannot accept the interpretation of the results. I can agree that cannabis is not for the PCPs, and I do maintain the arguments for the possible use of cannabis derivatives by the palliative care specialists. However, I am not an enthusiast of marijuana at all, as you can see.

Please, note, that opioids are much more dangerous drugs than THC, especially in hands of unexperienced physician.

- lines 538-540: it is not supported by evidence and national guidelines / recommendations as well as most recent EFIC (European Pain Federation) position paper on appropriate cannabinoids use.

Answer: This is my interpretation of the analyzed data. But you are right, and I added an appropriate statement supporting your remark in lines 554-557.

Reviewer 2 Report

Dr. DzierĹĽanowski has provided a thorough overview of cannabinoids, both phytocannabinoids as well synthetic cannabinoids, and briefly described their mode of action via CB1 and CB2 receptors in various organ systems. The review article also highlights the drug metabolism, interaction, tolerance and safety issues associated with cannabinoids, particularly in the context of medical use. Finally, a brief section helps us appreciate the application of cannabinoids in cancer treatment. While providing an excellent overview of these various subtopics, the review article also touches on the non-canonical interactions of cannabinoids with receptors other than CB1 and CB2 e.g. PPAR gamma, GPR18 etc. It is this promiscuity of receptor interaction and coupling with several major downstream pathways, including metabolism, that is successfully being explored in vitro & in animal models to better understand the various mechanisms by which cannabinoids exert anti-tumor activity. Therefore this phenomenon warrants a more in-depth discussion in the section pertaining to cannabinoids and cancer. Overall, articles such as the present review, can help inform the field as well public opinion and therefore health on the safety, efficacy, successes and potential of cannabinoids in medical as well as recreational use.

Author Response

Thank you very much for your fair and rather generous feedback.

I considered to write more on anti-cancer mode of action of cannabis, but finally I realized that it would double the size of the paper, and what more, it would focus the reader on possible anti-cancer use of marijuana instead of the evidence based symptomatic use.

I think the best solution would be a separate article on cannabis as a means of cancer treatment.

I have entered some changes in the text, though. Here is the list of the lines with corrections:

reference number: 312, 360, 393, 452, 476, 477, 482, 489, 496, 516

grammar/interpunction 124, 153, 247, 262, 274-276, 313, 559-561

format 317

style/text change: 225-234, 354-356, 518-530, 541-542, 547, 549, 554-557

Once again, thank you very much.
